

# UFS-RAQMS Global Atmospheric Composition Model: TROPOMI CO Column Assimilation

Maggie Bruckner[1], R. Bradley Pierce[1,2], Allen Lenzen[2], Glenn Diskin[3], Josh DiGangi[3], Martine De Maziere[4], Nicholas Jones[5], Maria Makarova[6]

[1]Department of Atmospheric and Oceanic Sciences, University of Wisconsin-Madison, Madison, WI, 53706, USA
[2]Space Science and Engineering Center, University of Wisconsin–Madison, Madison, WI, 53706, USA
[3]NASA Langley Research Center, Hampton, VA, 23681, USA
[4]Royal Belgian Institute for Space Aeronomy (BIRA-IASB), Brussels, 1180, Belgium
[5]School of Chemistry, University of Wollongong, Wollongong, Australia
[6]Atmospheric Physics Department, Saint Petersburg State University, Saint Petersburg, Russia

*Correspondence to*: R. Bradley Pierce (rbpierce@wisc.edu)

## Abstract

This paper describes a new version of the Real-time Air Quality Modeling System (RAQMS) which uses National Unified Operational Prediction Capability (NUOPC) coupling to combine the RAQMS chemical mechanism with the Global Ensemble Forecasting System with Aerosols (GEFS-Aerosols), the Goddard Chemistry Aerosol Radiation and Transport model (GOCART) aerosol mechanism, and NOAA's Unified Forecast System (UFS) version 9.1 Finite Volume Cubed Sphere (FV3) dynamical core. We also present an application of TROPOMI CO column data assimilation in UFS-RAQMS with the NOAA Grid Point Statistical Interpolation (GSI) three-dimensional variational (3Dvar) analysis system to constrain UFS-RAQMS CO. We validate UFS-RAQMS control and TROPOMI CO data assimilation CO analyses for the period 15 July – 30 September 2019 against independent satellite, ground based, and airborne observations. We show the largest impacts of the TROPOMI CO data assimilation are in the lower troposphere over Siberia and Indonesia. We find UFS-RAQMS biomass burning signatures in CO column are not consistent with those in AOD near the Siberian and Indonesian biomass burning source regions within our control experiment. Assimilation of TROPOMI CO improves the representation of the biomass burning AOD/CO relationship. The results also indicate that the biomass burning CO emissions from the Blended Global Biomass Burning Emissions Product (GBBEPx) used in UFS-RAQMS are too low.

## 1. Introduction

The Real-time Air Quality Modeling System (RAQMS) is a global chemical transport model with full stratospheric and tropospheric chemistry (Pierce et al., 2007, 2009). We have incorporated the RAQMS unified stratosphere/troposphere chemistry, photolysis, and wet and dry deposition modules into NOAA's Unified Forecast System (UFS) to produce a global atmospheric composition assimilation and forecasting system hereafter referred to as UFS-RAQMS. In this study we demonstrate the impact of TROPOMI CO total column data assimilation in UFS-RAQMS utilizing the NOAA Grid Point Statistical Interpolation (GSI) three-dimensional variational (3Dvar) analysis system (Kleist et al., 2009; Wu et al., 2002).



Carbon monoxide (CO) is an important atmospheric trace gas due to both its influence on OH and ozone ($O_3$) chemistry and its use as a pollution transport tracer. The major loss pathway for CO is its reaction with OH (Logan et al., 1981), and this reaction significantly impacts the oxidizing capacity of the atmosphere. CO sources include production during VOC oxidation and direct emission from biomass burning and fossil fuel combustion. Chemical transport models (CTMs) frequently underestimate CO (eg. Naik et al., 2013; Shindell et al., 2006; Strode et al., 2015).

Potential reasons for this include underestimation of anthropogenic and/or biomass burning emissions, overestimation of OH, and underestimation of secondary CO production from VOCs.

    Biomass burning emissions inventories have a high uncertainty due to factors including the incomplete knowledge of the spatiotemporal distribution of sources and limitations in capturing variation in fuel and fire behavior characteristics (eg. Hyer and Reid, 2009; Pan et al., 2020). CTM forecasts vary significantly depending on which biomass burning

emission inventory is used (eg. Bian et al., 2007; Pan et al., 2020; Stockwell et al., 2022). Additionally, biomass burning emissions schemes use emission ratios relative to CO for determining the release of VOCs and other non-CO emissions further compounding the effect of poor biomass burning emissions on CTM forecast skill for VOC-NOx-$O_3$ chemistry.

    Chemical data assimilation (DA) systems can be used to reduce the impacts of emissions uncertainty and model

deficiencies in representing sub-grid scale processes by using atmospheric composition measurements to constrain CTM fields. Chemical DA capabilities have been developed by modifying meteorological DA systems to use chemical concentration measurements. DA methods implemented for chemical DA include optimal interpolation-based methods (eg. Lamarque et al., 1999; Lamarque and Gille, 2003; Pierce et al., 2007), 3D variational methods (Pagowski et al., 2010), and 4D variational methods (eg. Inness et al., 2015; Inness et al., 2022). Chemical DA improves the CTM

analysis through minimizing the difference between observations and model analyses. Observation datasets with a higher spatial coverage during the assimilation window provide more information about the true atmospheric composition. DA systems have been used to assimilate remote sensing observations of CO from Measurement of Air Pollution from Space (MAPS), Interferometric Monitor for Greenhouse Gases (IMG), Measurements of Pollution in the Troposphere (MOPITT), Infrared Atmospheric Sounding Interferometer (IASI), and TROPOMI (eg. Barré et al.,

2015; Clerbaux et al., 2001; Inness et al., 2015; Inness et al., 2022; Lamarque et al., 1999).

    In this study we introduce UFS-RAQMS and apply it to TROPOMI CO DA during July-August-September (JAS) 2019. During JAS the 2019 NASA/NOAA Fire Influence on Regional to Global Environments and Air Quality (FIREX-AQ) field campaign (Warneke et al., 2023) sampled smoke plumes over North America. The NASA Cloud, Aerosol and Monsoon Processes Philippines Experiment (CAMP²Ex) field campaign (Reid et al., 2023) occurred 25

August – 5 October 2019 and sampled smoke over the maritime continent. Fire activity in the continental US in 2019 was significantly below average, thought to be the result of higher fuel moisture content (Warneke et al., 2023). Globally, biomass burning emissions typically peak around August-September (van der Werf et al., 2017). The Blended Global Biomass Burning Emissions Product (GBBEPx) calculates daily biomass burning emissions using observations of fire radiative power (FRP) from Moderate Resolution Imaging Spectroradiometer (MODIS) (Aqua

and Terra satellites) and Visible Infrared Imaging Radiometer Suite (VIIRS) (Suomi NPP and NOAA-20 satellites)



(Zhang et al., 2019). Siberian wildfire emissions peaked during July and August 2019, and by September global biomass burning emissions were predominantly due to burning in the tropics (fig. 1). Smoke from the Siberian wildfires was transported over North America, where it impacted tropospheric composition and surface air quality (Johnson et al., 2021). Smoke from drought-enhanced biomass burning in the maritime continent contributed to

September 2019 having the 3rd highest AOD in the MODIS record, behind significant enhancements in 2006 and 2015 (Reid et al., 2023).

This paper is structured as follows. Section 2 describes the UFS-RAQMS model and data assimilation system as well as the method for obtaining the background error covariance statistics. Section 3 shows results from the TROPOMI CO DA experiment and validation with independent observations. Section 4 evaluates the relationship between CO

column and AOD for 2 case studies- one over Siberian biomass burning in July and one over Indonesia in September. Conclusions are given in section 5.

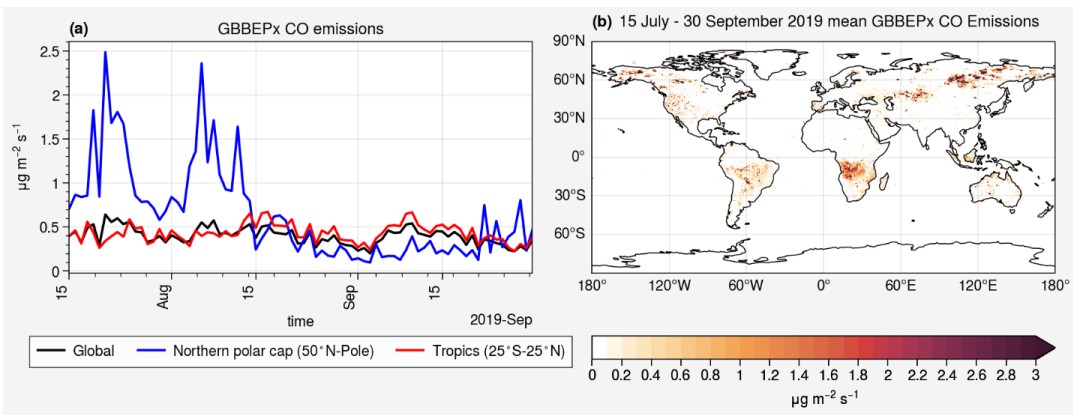

**Figure 1. GBBEPx emissions during 15 July – 30 September 2019.** Panel (a) displays time series of over selected regions. Panel (b) mean spatial distribution of GBBEPx emissions.

**2. UFS-RAQMS Model**

UFS-RAQMS is an updated version of the Real-time Air Quality Modeling System (RAQMS) (Pierce et al., 2007) where the RAQMS stratosphere/troposphere chemistry, photolysis, and wet and dry deposition modules are coupled to NOAA's Unified Forecast System (UFS) version 9.1 Finite Volume Cubed Sphere (FV3) dynamical core (Harris and Lin, 2013; Putman and Lin, 2007). The UFS-RAQMS configuration utilized in this study is an extension of the

operational NOAA Global Ensemble Forecasting System with Aerosols (GEFS-Aerosols, Bhattacharjee et al., 2023; Zhang et al., 2022). GEFS-Aerosols includes bulk aerosol modules from the Goddard Chemistry Aerosol Radiation and Transport model (GOCART, Chin et al, 2002).

The extension is accomplished by coupling RAQMS chemistry, photolysis, and wet and dry deposition modules with the UFS dynamical core through the National Unified Operational Prediction Capability (NUOPC,

https://earthsystemmodeling.org/nuopc/) layer. The NUOPC layer defines conventions and generic components for



building coupled models using the Earth System Modeling Framework (ESMF, https://earthsystemmodeling.org). This NUOPC based coupling allows the GOCART aerosol predictions to impact the RAQMS Fast-J2 (Bian and Prather, 2002) photolysis scheme and also allows the RAQMS OH and H2O2 predictions to impact the GOCART sulfate aerosol formation.

The RAQMS chemistry module utilizes a family approach to reduce the number of species considered in the chemical mechanism, requiring solving of continuity equations for 55 chemical families and constituents and determination of equilibrium concentrations for 86 separate species (Pierce et al., 2007). Non-methane hydrocarbon chemistry in the RAQMS chemistry module follows the lumped-structure Carbon Bond Mechanism Z (CB-Z) (Zaveri and Peters, 1999), which was modified in Pierce et al. (2007) to include an explicit isoprene oxidation scheme. Standard hydrogen
oxide (HOx), chlorine oxide (ClOx), bromine oxide (BrOx), and NOx ozone photochemistry (Eckman et al., 1995) is also included.

In this study we conduct UFS-RAQMS retrospective simulations during July 15, 2019 through September 30, 2019 at a Cubed Sphere resolution of 192 (C192, 192x192 grid-points within each 6 cubes or approximately 0.5° x 0.5° horizontal resolution) with 64 hybrid vertical levels from the surface to upper stratosphere (approximately 0.2hPa).
The UFS-RAQMS atmospheric composition experiments are conducted in "replay" mode, with UFS-RAQMS meteorological fields are initialized with GFS analyses at 6-hour intervals followed by UFS-RAQMS forecasts with and without data assimilation cycling. The forecasts without data assimilation are used as the control experiment. Both UFS-RAQMS experiments were initialized on 15 July 2019 at 12Z with 1x1 degree analyses from RAQMS, which includes assimilation of NASA Moderate Resolution Imaging Spectroradiometer (MODIS) AOD on the Terra and
Aqua satellites and the NASA Ozone Monitoring Instrument (OMI) cloud cleared total column ozone and Microwave Limb Sounder (MLS) stratospheric ozone profiles. Global anthropogenic emissions in UFS-RAQMS are obtained from the Community Emissions Data System (CEDS, McDuffie et al., 2020). Daily global biomass burning CO emissions are specified from GBBEPx (Zhang et al., 2019.) and is expanded for other trace gas emissions using species specific emissions factors from the RAQMS biomass burning scheme (Soja et al., 2004). Fire Radiative Power (FRP)
is used to calculate GBBEPx plume rise (Ahmadov et al., 2017).

**2.1 GSI 3D-Var**

This study uses the operational grid point statistical interpolation (GSI) 3D variational (3DVAR) DA system (Kleist et al., 2009; Wu et al., 2002) to assimilate TROPOMI CO columns. Within this implementation, the UFS-RAQMS 3D CO volume mixing ratio is used as the analysis variable in the minimization procedure. The background error
covariance (BEC) statistics for CO are obtained using the National Meteorological Center (NMC) method (Descombes et al., 2015; Parrish and Derber, 1992). The NMC method typically uses differences between 24-hour forecasts and 48-hour forecasts to estimate BEC statistics. Here, in addition to the standard BEC implementation, we apply the NMC method to a pair of forecasts that have different biomass burning emissions to account for uncertainties in CO emissions. The biomass burning emission BEC statistics are computed from the differences between 100% GBBEPx
CO emissions and 85% GBBEPx CO emissions UFS-RAQMS CO forecasts. The biomass burning and forecast BEC





statistics are combined in a piecewise-linear fashion to create "blended" BEC statistics. We set the blended BEC statistics equal to the standard, forecast-sensitive BEC statistics above model level 25 (approximately 480hPa). Below model level 15 (approximately 780hPa), the blended BEC statistics are equal to the biomass burning BEC statistics with an inflation factor of 5 applied to the standard deviation. Between model levels 15 and 25, the two BEC estimates

are linearly blended.

### 2.1.2 TROPOMI CO Total Column

The Tropospheric Monitoring Instrument (TROPOMI) (Veefkind et al., 2012) is a higher resolution follow-on to the NASA Ozone Monitoring Instrument (OMI) that is currently in orbit on-board ESA's polar-orbiting Sentinel-5 precursor satellite that observes in UV-near IR and shortwave IR. We use the v2.4.0 CO total column retrieval with

the striping correction applied (Borsdorff et al., 2019). Following recommended quality assurance guidelines (**https://sentinel.esa.int/documents/247904/3541451/Sentinel-5P-Carbon-Monoxide-Level-2-Product-Readme-File.pdf**, last access: 18 July 2024), we use observations with a quality assurance value 1 (best) over land and 0.7 (OK, but mid-level clouds present) over ocean. This leads to assimilation of only cloudy data over ocean, as the clear sky ocean retrieval signal intensity is too weak (Inness et al., 2022).

TROPOMI CO has a spatial resolution of 5.5 x 3.5 km (7 x 3.5km prior to August 6, 2019), which is higher than UFS-RAQMS resolution. Owing to this difference in resolution, multiple TROPOMI observations may fall within a model grid cell during the assimilation window. Unlike other studies that utilize satellite CO "super-observations" (eg. Gaubert et al., 2020; Inness et al., 2022; Sekiya et al., 2021), we assimilate observations individually since using super-observations smooths the spatial variability in analysis increments (Sekiya et al., 2021). This smoothing could lead to

underestimates in localized CO column enhancements associated with biomass burning.

Figure 2a shows the mean TROPOMI CO columns over the continental US during the FIREX-AQ field campaign and the NASA DC-8 flight tracks. CO columns over the central and eastern US are ~2x higher than over the western US largely due to higher topography in the western US and thus thinner atmospheric columns. Figure 2b shows the mean TROPOMI CO concentrations over SE Asia during the CAMP$^2$Ex field campaign and the NASA P-3 flight tracks.

During CAMP$^2$Ex high CO columns (>$4 \times 10^{18}$ mol/cm$^2$) over the islands of Borneo and Sumatra are due to the sustained burning of peatlands (Reid et al., 2023).





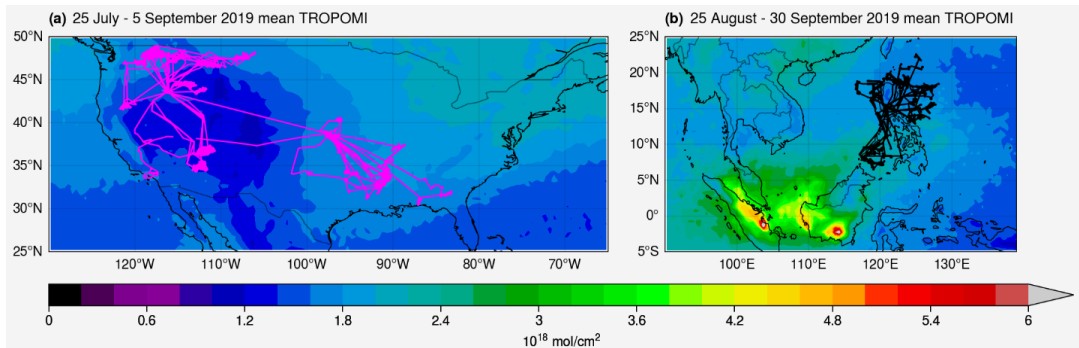

**Figure 2. Mean TROPOMI CO columns over the US (a) and SE Asia (b). FIREX-AQ DC-8 flight tracks (pink) and CAMP$^2$Ex P-3 (black) flight tracks are shown over the respective campaign domains.**

**3. Impact of TROPOMI CO Assimilation on UFS-RAQMS CO**

The UFS-RAQMS control CO columns are lower than the TROPOMI CO column observations in the NH and higher in the SH (figure 3). Figure 3 also shows the FIREX-AQ and CAMP2Ex field campaign domains and the locations of Network for the Detection of Atmospheric Composition Change (NDACC) Fourier-transform infrared (FTIR) spectrometers used to validate UFS-RAQMS CO profiles. NDACC is a global network consisting of 80 currently

active stations providing high quality observations of atmospheric trace gases and aerosols using ground-based insitu- and remote sensing techniques including ozonesondes, FTIR spectrometers, lidar, and UV/visible spectroscopy (De Mazière et al., 2018). The UFS-RAQMS control experiment significantly underpredicts CO columns over central Africa, the maritime continent, and Siberian Russia. Figure 1b shows that each of these regions are associated with significant biomass burning during this period.

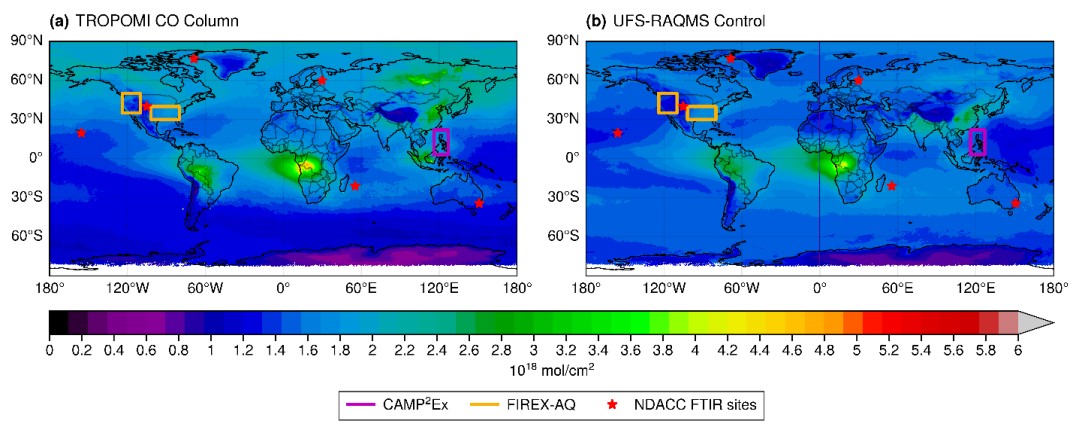


**Figure 3. 15 July- 30 September 2019 average CO Column concentrations for (a) TROPOMI and (b) UFS-RAQMS control. Boxes indicate domains for CAMP$^2$Ex (purple) and FIREX-AQ (yellow) campaigns. NDACC FTIR locations utilized in this study are denoted by red stars.**



### 3.1 Differences in CO between control and DA experiments

To quantify the impact of assimilating TROPOMI CO on UFS-RAQMS analyses, we calculate the average percent change in zonal mean CO and CO total column between the control and TROPOMI CO DA experiments. Figure 4a shows that the assimilation increases tropospheric zonal mean CO north of 20°S and decreases zonal mean CO above the tropopause. Above the tropopause the largest impact of the TROPOMI CO DA on CO is a decrease of 32-52% in the southern hemisphere (SH) between 40°S and 60°S and 11-13 km. The stratospheric regions with the largest

decreases are in the midlatitudes and characterized by a strong vertical gradient in CO that sharpens as a result of the TROPOMI CO DA. This stratospheric percentage change is associated with low CO concentrations. These large stratospheric differences are not a direct consequence of the TROPOMI assimilation, as zonal mean cross sections of the analysis increments (not shown) illustrate that the DA primarily adjusts CO in the troposphere. Stratospheric CO analysis increments are concentrated near the tropopause and largest in the polar NH. Consequently, these large SH

stratospheric CO percentage changes most likely arise from reductions in CO in the tropical upper troposphere through TROPOMI CO DA and then cross tropopause transport of the lower CO from the tropical upper troposphere into the stratosphere.

The largest increases in zonal mean CO concentrations are between 45° N and 80°N below 5km and in excess of 60%. Figure 4b shows that the assimilation tends to increase CO total column north of 30°S and decrease CO total column

south of 30°S. The largest increases in CO total column are in excess of 60% and in Siberia and the maritime continent, which during this time period experienced significant biomass burning activity.



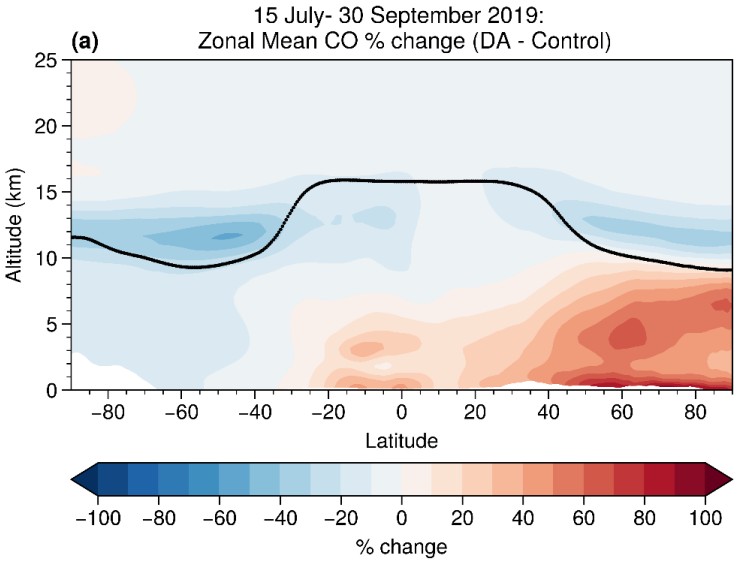

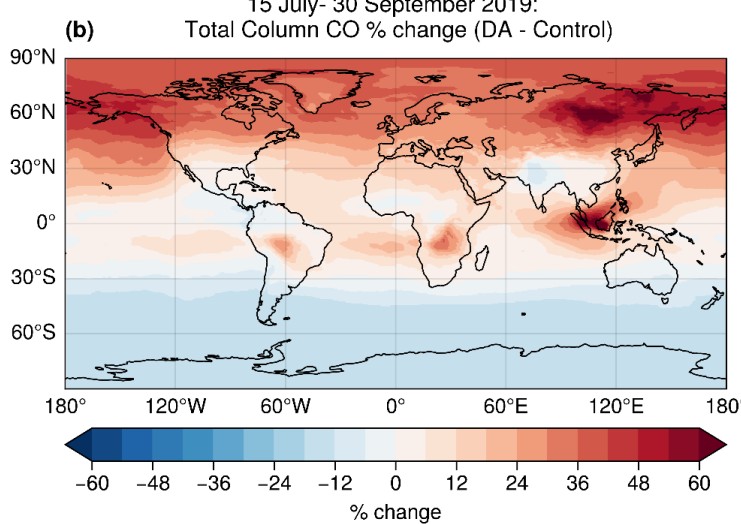

**Figure 4. Percent difference in zonal mean CO profile (a) and total column CO (b) between UFS-RAQMS TROPOMI CO DA and control experiments.**

**3.2 Evaluation with independent datasets**

We evaluate improvement in UFS-RAQMS CO due to TROPOMI CO DA through validation with independent observations from the Measurements of Pollution in the Troposphere (MOPITT) instrument, the NASA/NOAA FIREX-AQ field campaign, the CAMP2EX field campaign, and NDACC FTIR spectrometers. The control and TROPOMI CO DA experiments are spatially and temporally interpolated to the observation, creating coincident





model and observation pairs. For the MOPITT and NDACC comparisons, we apply the observation averaging kernels to the UFS-RAQMS coincident profiles.

**3.2.1 MOPITT CO Column**

We compare daily mean UFS-RAQMS total column CO analyses with the MOPITT version 9 Level 3 daily mean CO column product (Deeter et al., 2022). Due to an event upset affecting instrument operation MOPITT data is

unavailable    for    a    large    portion    of    the    study    period    (26    July    -    24    August    2019) (https://www2.acom.ucar.edu/mopitt/status). Daily average UFS-RAQMS CO is binned onto the MOPITT level 3 grid, then linearly interpolated to the MOPITT vertical levels. The average daily MOPITT CO column for 15 July - 30 September 2019 is shown in figure 5a.  A root mean square error (RMSE) skill score (equation 1) is used to quantify the improvement in the TROPOMI CO DA experiment. The RMSE for UFS-RAQMS control and UFS-RAQMS

TROPOMI CO DA are calculated relative to the MOPITT observations. Negative skill scores indicate that the assimilation degraded the forecast while positive skill indicates the assimilation increased the accuracy of the forecast. A skill score of 1 indicates that the TROPOMI CO DA experiment captures the CO columns as depicted by MOPITT. A skill score of 0 indicates that the assimilation did not improve the agreement between MOPITT and UFS-RAQMS or that the model has no skill in capturing CO in that region.

$$RMSE\_SS_{(i,j)} = 1 - \frac{RMSE_{DA(i,j)}}{RMSE_{ctrl(i,j)}} \qquad (1)$$

For most grid cells the UFS-RAQMS TROPOMI CO DA experiment exhibits improved skill (fig 5c). The largest improvements in skill are over Russia, Europe, Alaska, and Canada. Due to the MOPITT data outage, the large Siberian biomass burning events are not captured within the MOPITT observations except for in the first 10 days of the experiment. Therefore, while we are unable to directly verify the increased UFS-RAQMS CO within the Siberian

smoke plume during August 2019, we do show that assimilating TROPOMI CO throughout the period significantly improved background CO concentrations in the NH middle and high latitudes. UFS-RAQMS TROPOMI CO DA analyses also show improvement over regions of Africa and the maritime continent where there was widespread biomass burning during the analysis period. TROPOMI CO DA results in slight improvements over the Pacific Ocean and negative skill in the eastern Tropical Pacific near the coast of Mexico.

In addition to the RMSE skill score, we compare the daily mean UFS-RAQMS CO column analyses with MOPITT CO columns over the FIREX-AQ and CAMP$^2$Ex field campaign domains in figure 6. Correlation and bias are calculated between all observations made 15 July-30 September 2019 over 30°N - 49.5°N 82°W - 123°W (fig. 6 a,b) and 6°N - 23°N 116°E - 129°W (fig. 6 c,d). Over the FIREX-AQ domain, TROPOMI CO DA increases correlation of UFS-RAQMS with MOPITT from 0.661 to 0.8317 and decreases the bias from -0.2507 x10$^{18}$ mol/cm$^2$ to -0.0354

x10$^{18}$ mol/cm$^2$. Over the CAMP$^2$Ex domain, TROPOMI CO DA increases correlation of UFS-RAQMS with MOPITT from 0.495 to 0.9446 and decreases the bias from -0.3114 x 10$^{18}$ mol/cm$^2$ to 0.1437 x 10$^{18}$ mol/cm$^2$.



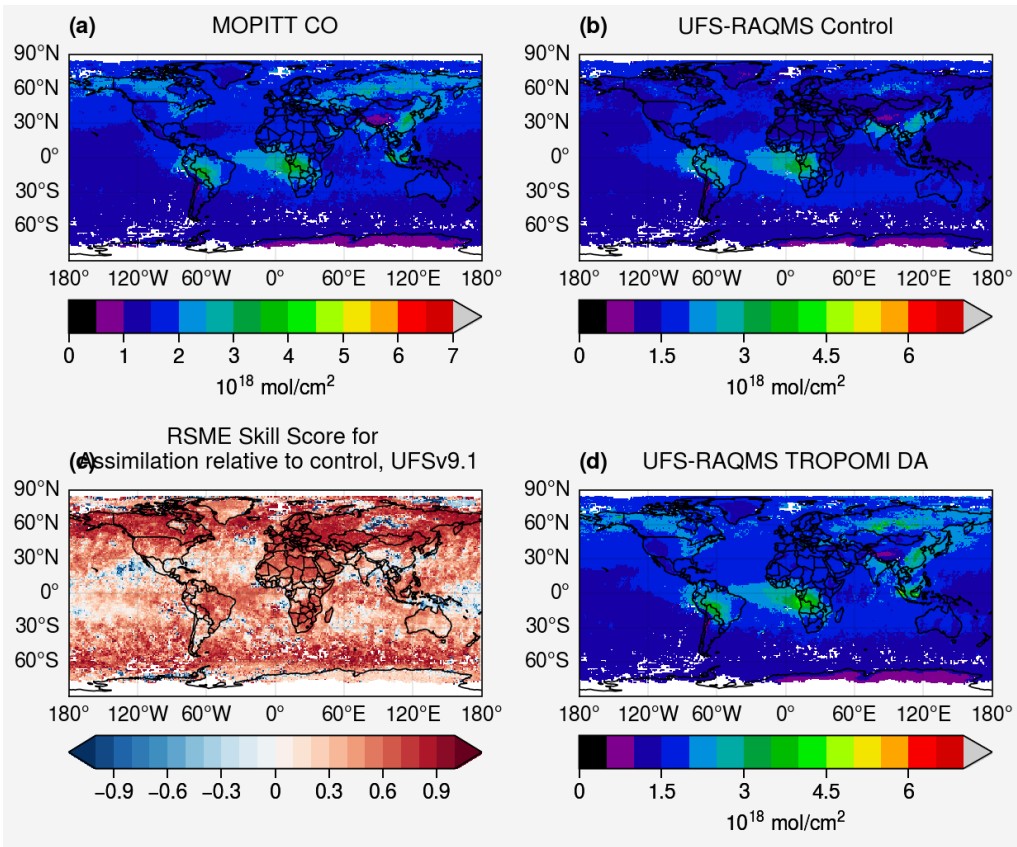

**Figure 5. Comparison of MOPITT CO Column with UFS-RAQMS control and TROPOMI CO DA CO columns. 15 July-30 September 2016 mean CO column for MOPITT (a), UFS-RAQMS control (b), and UFS-RAQMS TROPOMI CO DA (d), with 26 July - 24 August 2019 excluded due to MOPITT data outage. RMSE Skill Score (c) shows improved agreement with MOPITT in UFS-RAQMS TROPOMI CO DA over UFS-RAQMS control.**




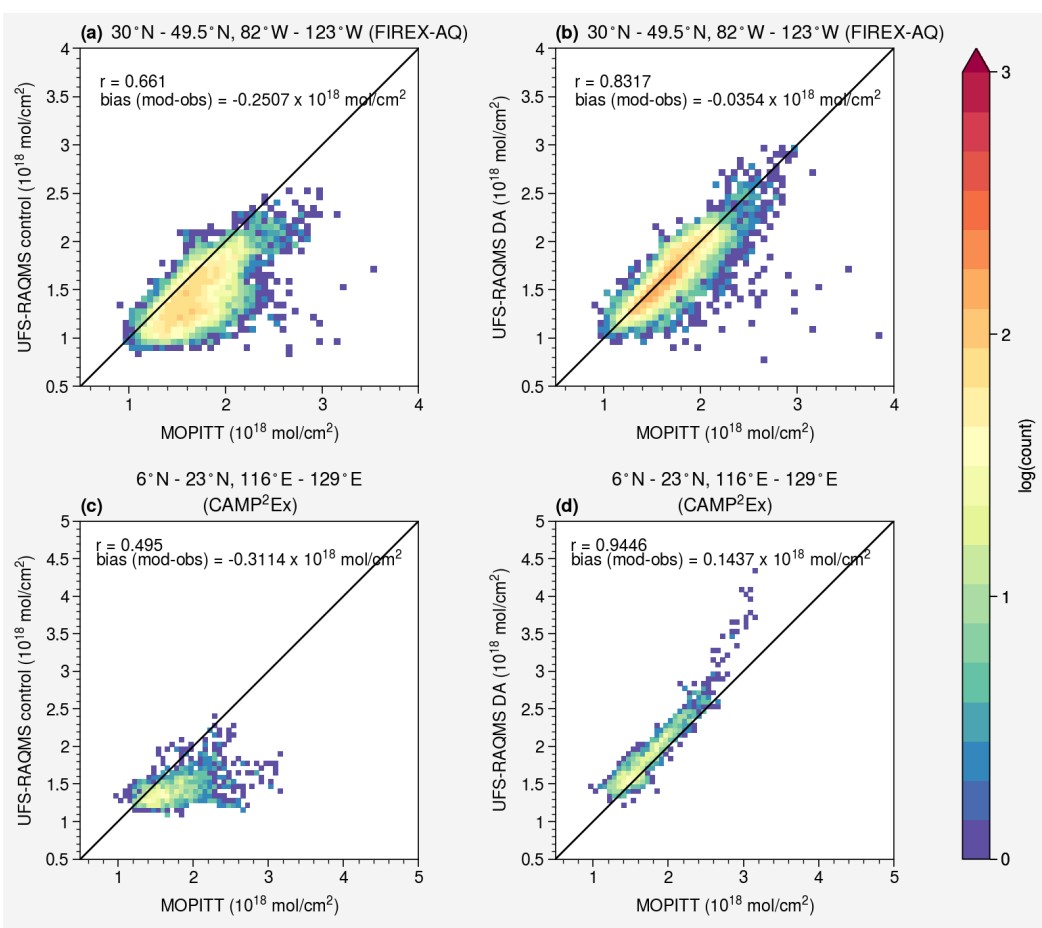

**Figure 6. Comparisons of MOPITT and UFS-RAQMS CO Columns over FIREX-AQ (a,b) and CAMP²Ex (c,d) domains. TROPOMI CO DA increases correlation and decreases bias between UFS-RAQMS and MOPITT.**

**3.2.2 FIREX-AQ In-situ measurements**

Over the continental US from July-September the Differential Absorption Carbon Monoxide Measurement (DACOM) instrument (Sachse et al., 1991) made measurements onboard the NASA DC-8 as part of the FIREX-AQ field campaign. FIREX-AQ sampling of smoke plumes with the DC-8 consisted of multiple perpendicular transects through the plume, with each perpendicular leg sampling smoke emitted around the same time, and the legs starting in the

freshest smoke (Warneke et al., 2023). The resulting FIREX-AQ highly detailed measurements capture fine-scale changes in composition in both the cross-plume direction and as the emissions age. In the following comparisons, in-plume measurements are excluded from the analysis as the horizontal resolution of the UFS-RAQMS simulations is not fine enough to capture the observed in-plume enhancements that were measured by the DC-8 close to the western US wildfires and SE US agricultural fires targeted during FIREX-AQ.




Figure 7 shows the comparison between UFS-RAQMS and the DC-8 DACOM CO observations for non-smoke plume observations during all flights during FIREX-AQ. UFS-RAQMS CO is strongly correlated with the observed CO for both the control (0.7956) and the TROPOMI CO DA experiment (0.8129). TROPOMI CO DA improves the average bias from -9.6635 ppbv to 6.2821 ppbv.

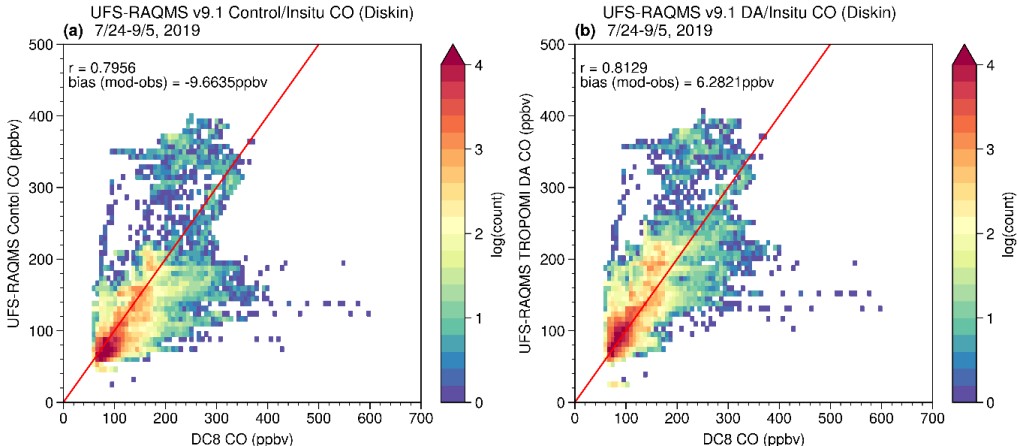

**Figure 7. Comparison of DC-8 DACOM CO and (a) UFS-RAQMS Control experiment and (b) UFS-RAQMS TROPOMI CO DA experiment.**


Figure 8 shows a comparison of the vertical profiles for the FIREX-AQ DACOM CO non-smoke observations and coincident UFS-RAQMS analyses. Following the interpolation of the UFS-RAQMS analyses along the DC-8 flight track and filtering out smoke observations, the modeled and measured values were binned into 200 m altitude bins.

The median (vertical profile), 25th and 75th (shaded) percentiles of the modeled and observed distributions within each 200m altitude bin are shown. Below 2 km the control and TROMPOMI CO DA experiment profiles are both within the spread for the observed profile. Above 2.5 km the control experiment profile is consistently biased low relative to the observed profile. The TROPOMI CO DA experiment profile is higher than in the control experiment and show improved agreement with the DC-8 observations.



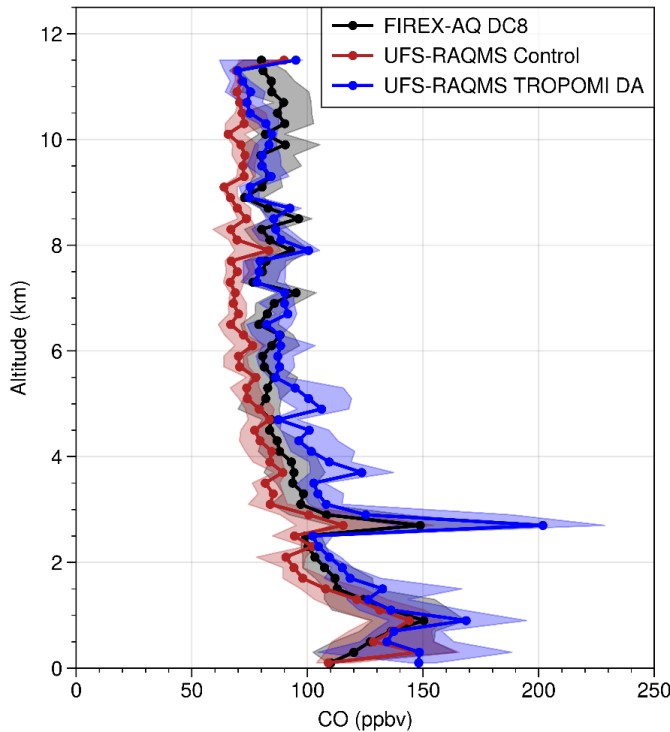

**Figure 8. Vertical profiles of non-smoke CO during FIREX-AQ for DC-8 DACOM CO (black), UFS-RAQMS Control experiment (red), and UFS-RAQMS TROPOMI CO DA experiment (blue).**

### 3.2.3 CAMP²Ex In-situ measurements

The NASA CAMP²Ex field campaign sampled airmasses over the Philippines 25 August–5 October 2019 with the NASA P-3 aircraft to investigate the role of aerosols in the Southeast Asian southeast monsoon (Reid et al., 2023). During the campaign, the region was impacted by significant biomass burning. In-situ CO measurements were made by a commercial cavity ringdown spectrometer (G2401-m, PICARRO, Inc.) modified with a custom gas sampling system (DiGangi et al., 2021). UFS-RAQMS analyses are sampled along the P-3 flight track. Figure 9 shows the comparison between UFS-RAQMS and the CAMP²Ex P-3 CO measurements. The UFS-RAQMS Control experiment has a low bias of -34.553 ppbv relative to the observations and is moderately correlated (0.7332). Assimilating TROPOMI CO decreases the bias in the analysis significantly to -1.8373 ppbv and improves the correlation (0.8202).





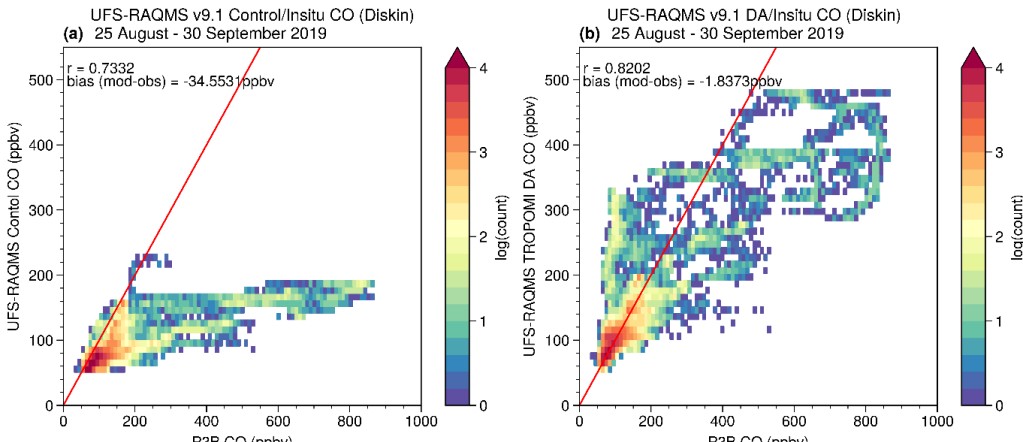

**Figure 9. Comparison of CAMP²Ex P3-B CO and (a) UFS-RAQMS Control experiment and (b) UFS-RAQMS TROPOMI CO DA experiment.**

Figure 10 shows a comparison of the vertical profiles for the CAMP²Ex CO observations and coincident UFS-RAQMS analyses. Following the interpolation of the UFS-RAQMS analyses along the P-3 flight track, the modeled and measured values were binned into 200 m altitude bins. The median (vertical profile), 25th and 75th (shaded) percentiles of the modeled and observed distributions within each 200m altitude bin are shown. Below 7km, the UFS-RAQMS control experiment profile is biased low by ≥ 20 ppbv (≥ 20%) relative to the observed profile. This low

bias is largest in the lowest 1.5km where it exceeds -40%. The UFS-RAQMS TROPOMI CO DA experiment profile is generally within the 25th-75th percentiles of the CAMP²Ex observations, though between ~3.5km and 5km the UFS-RAQMS DA CO profile is biased high and may indicate a slight overcorrection. The lowest 1km of the profile is still biased low, though it is now only 10-20%.

The comparisons of UFS-RAQMS to the in-situ FIREX-AQ and CAMP²Ex observations show that TROPOMI CO

DA improves the correlation and bias statistics. This is consistent with the MOPITT statistics over the campaign domains. However, the improvement in the statistics is better for UFS-RAQMS CO columns than for the UFS-RAQMS CO profiles. This is a result of using a total column measurement to constrain the UFS-RAQMS CO analysis. UFS-RAQMS control CO is strongly correlated with the in-situ observations, indicating along with the profiles that the vertical structure and temporal variation in CO concentration is reasonably captured in UFS-RAQMS for these

regions. The DA system distributes the analysis increment vertically based on model blended BEC statistics and knowledge of observation errors and vertical sensitivities. Over the CAMP²Ex domain this leads to an overestimation of CO at 3-6km. Over the FIREX-AQ domain this leads to an overestimate of CO below 6km and underestimates above 10km. In the UFS-RAQMS TROPOMI CO DA experiment column, the effects of the vertical distribution compensate for each other.



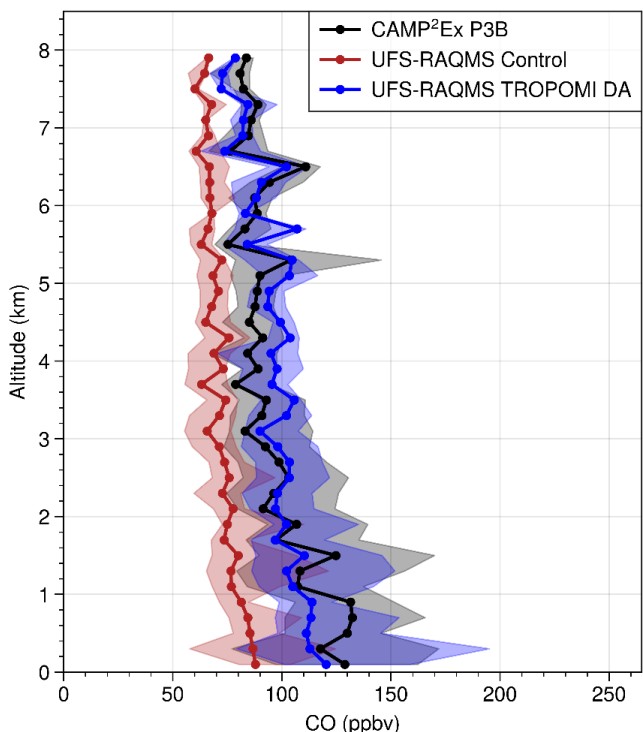


**Figure 10. Vertical profiles of CO during CAMP²Ex for P-3 CO observation (black), UFS-RAQMS Control experiment (red), and UFS-RAQMS TROPOMI CO DA experiment (blue).**

### 3.2.4 NDACC FTIR

UFS-RAQMS CO profiles are also evaluated with FTIR CO profile observations from 6 NDACC sites (table 1). The

selected NDACC FTIR spectrometers retrieve volume mixing ratio profiles from solar absorption spectra with optimal

estimation using the SFIT4 algorithms (https://wiki.ucar.edu/display/sfit4/, last access: 19 July 2024).



Table 1. Location of NDACC FTIR sites used in this study. Number of profiles taken 15 July- 30 September 2019 included.

| NDACC Site Name | Number of Profiles | Location (Latitude/Longitude) |
|---|---|---|
| Boulder, CO, USA | 288 | 39.99ºN, 105.26ºW |
| La Reunion, Maido, France | 531 | 21.1ºS, 55.4ºE |
| Mauna Loa, HI, USA | 54 | 19.54ºN, 155.58ºW |
| St. Petersburg, Russian Federation | 76 | 59.9ºN, 29.8ºE |
| Thule, Greenland | 655 | 76.53ºN, 68.74ºW |
| Wollongong, Australia | 263 | 34.41ºS, 150.88ºE |


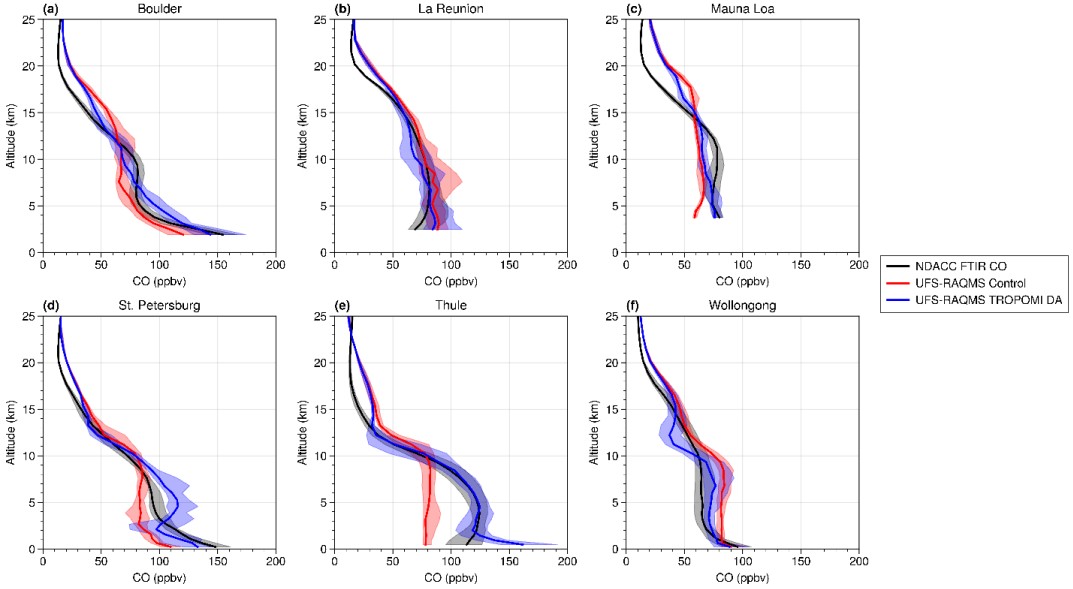

**Figure 11. Comparison of CO profiles from NDACC FTIR (black), UFS-RAQMS control (red), and UFS-RAQMS DA (blue). Solid lines indicate the median, shading 25th-75th percentile.**



UFS-RAQMS analyses were paired to the NDACC FTIR locations using a nearest-neighbor approach in the horizontal
followed by linear interpolation in the time and vertical dimensions. NDACC FTIR averaging kernels are then applied
to the UFS-RAQMS profiles. Figure 11 shows a comparison of NDACC FTIR CO profiles with UFS-RAQMS. The
influence of TROPOMI CO DA on the CO profile is small above 15km, with both the control and the TROPOMI CO
DA experiment generally overestimating CO concentrations in this region. The most significant differences between
the control and TROPOMI CO DA experiments occur below 10km except for at Wollongong where the most
significant difference is at 11-12km (fig. 11f). The Wollongong site is at 34.41ºS, 150.88ºE, where the mean impact
of the DA is a 20-30% decrease in CO (section 3.1, fig 4). At Wollongong, the TROPOMI CO DA reduces the average
high bias by 5-15 ppbv from 1-5km and ~10 ppbv from 5-10km but creates a low bias of ~15-20 ppbv from 10-12km.

Consistent with the percent change in CO between the control and TROPOMI CO DA experiments at high latitudes
in fig. 4, the Thule profile shows a significant increase in the profile due to the TROPOMI CO DA and results in very
good agreement with the observed NDACC profile from 2-13km. At Thule the near-surface CO concentration is
biased high in the TROPOMI CO DA experiment while it is biased low in the control. This behavior is not apparent
at the other sites and may be a consequence of the use of static BEC at these latitudes. Recall, the BEC statistics
obtained by this study are a function of latitude and altitude, and in the lower troposphere reflect the sensitivity of
UFS-RAQMS to biomass burning emissions. Profiles of the analysis increments at NDACC locations on the days that
measurements were made (not shown) indicate that the near-surface analysis increment is comparatively large
(>~15ppbv) at Boulder, St. Petersburg, and Thule. For Boulder and St. Petersburg, it appears the TROPOMI CO DA
is able to correct CO for biases in anthropogenic emissions since these sites were not significantly impacted by
wildfires.

At the tropical NDACC sites of Mauna Loa and La Reunion changes are small. TROPOMI CO DA slightly decreases
UFS-RAQMS CO at La Reunion and increases it below 15km at Mauna Loa.

**4 Consistency in biomass burning CO and aerosol signatures**

A strong relationship between black carbon aerosols and CO has been observed in airmasses dominated by biomass
burning emissions (eg. Arellano Jr. et al., 2010; Spackman et al., 2008) due to their co-emission during combustion.
Similarly, satellite aerosol optical depth (AOD) and CO column observations are strongly correlated over regions
where biomass burning is the dominant contributor to fine mode AOD (eg. Bian et al., 2010; Edwards et al., 2004,
2006). The correlation in space and time between AOD and CO is stronger in the southern hemisphere, while in the
NH peak AOD and CO loadings are offset due to the higher anthropogenic pollutant loading (Bian et al., 2010;
Buchholz et al., 2021; Edwards et al., 2004). Due to the shorter lifetime of biomass burning aerosols, enhancements
in AOD are a strong indicator of biomass burning emissions sources while CO is a good tracer of down-wind impacts
of those emissions due to its longer lifetime (eg. Bian et al., 2010; Buchholz et al., 2021; Edwards et al., 2006).
Edwards et al. 2006 also finds that the correlation between CO and AOD is strongest during the first few days of a
biomass burning event and declines as the local CO concentration becomes less representative of daily emissions.





Here, we evaluate the relationship between AOD and CO over two biomass burning events. VIIRS AOD and TROPOMI CO are used to evaluate how realistic the UFS-RAQMS AOD/CO relationship is. We selected scenes over

Siberia and over Indonesia during their respective peaks in biomass burning during the July-September 2019 analysis period.

UFS-RAQMS CO and AOD analyses are interpolated in latitude, longitude, and time to TROPOMI and VIIRS L2 observations respectively. TROPOMI averaging kernels are applied to UFS-RAQMS CO profiles. UFS-RAQMS speciated aerosol extinction profiles at 532nm are integrated to obtain AOD. The coincident model and observation

data is then binned onto a 0.1x0.1 degree grid. The anticipated linear relationship between AOD and CO is evaluated for the observations, UFS-RAQMS control, and UFS-RAQMS TROPOMI CO DA.

**4.1 Case Study: 22 July 2019 Siberian Smoke**

During July and August 2019 significant wildfire activity occurred in Siberian Russia, with a major cluster in Eastern Siberia and a major cluster in Central Siberia (Johnson et al., 2021). Wildfire activity peaked in both regions of Siberia

between 18 July and 26 July. We evaluate binned AOD and CO column on 22 July 2019 for the region 90°E -150°E, 50°N - 70°N.

The spatial distributions of AOD and CO over Siberia on 22 July 2019 are shown in figure 12 for VIIRS, TROPOMI, the UFS-RAQMS control, and the UFS-RAQMS TROPOMI CO DA experiment. The UFS-RAQMS AOD field is unchanged between the control and TROPOMI CO DA experiments and thus is only shown once. UFS-RAQMS does

a very good job of capturing the observed synoptic scale features but does not capture fine-scale structure seen in the AOD or CO observations. UFS-RAQMS AOD is slightly overestimated outside of the plume (AOD ≥ 1) and in the plume feature around 60°N - 70°N, 120°E - 130°E. CO column is significantly underestimated in UFS-RAQMS control. Agreement with the TROPOMI observations is significantly improved in UFS-RAQMS TROPOMI CO DA.



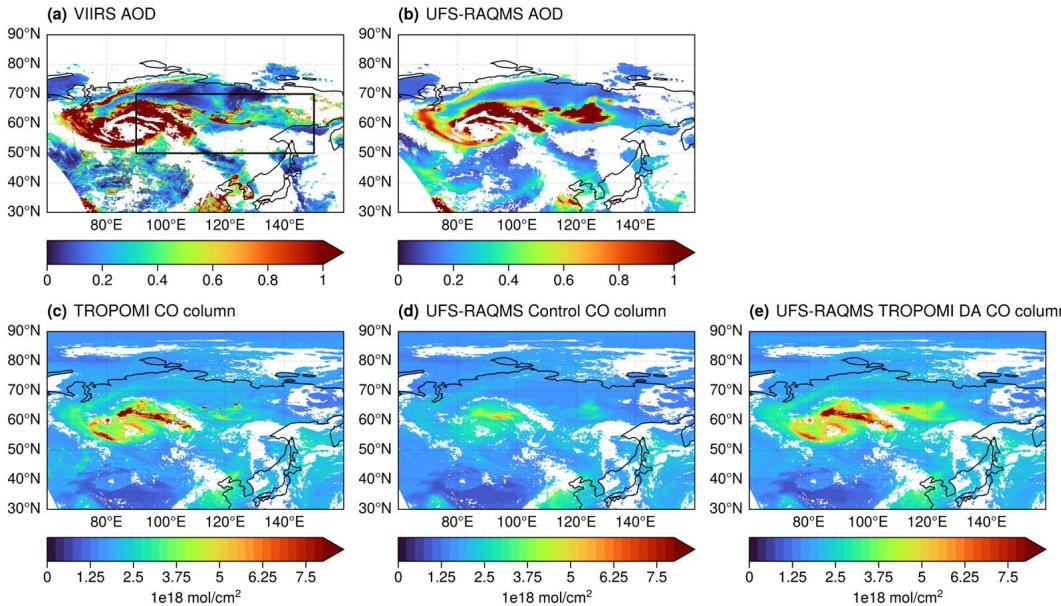

**Figure 12. 22 July 2019 AOD and CO columns over Siberia. VIIRS AOD (a), UFS-RAQMS AOD (b), TROPOMI CO column (c), and UFS-RAQMS control (d) and TROPOMI CO DA (e) CO column. Black box in panel a defines region (90°E -150°E, 50°N - 70°N) for AOD/CO column relationship analysis.**

Scatterplots illustrating the relationship between AOD and CO column in Siberian wildfire smoke are shown in figure 13 for the observations (grey), UFS-RAQMS control (red), and UFS-RAQMS TROPOMI CO DA (blue). The linear regressions are summarized in table 2. VIIRS AOD and TROPOMI CO Column exhibit a compact linear relationship with a slope near 1 and correlation of 0.8043. UFS-RAQMS control CO column and AOD are moderately correlated (0.5648), and the slope of the linear relationship is 0.2407 as UFS-RAQMS control underestimates of CO column for high AOD. TROPOMI CO DA improves the correlation between AOD and CO Column as well as increases the slope of the linear relationship. The UFS-RAQMS TROPOMI CO DA AOD/CO Column slope is 0.7749 and the correlation is 0.7106. This improved representation of the observed linear relationship and correlation in UFS-RAQMS TROPOMI CO DA is due to the increased CO column within the Siberian wildfire plume.



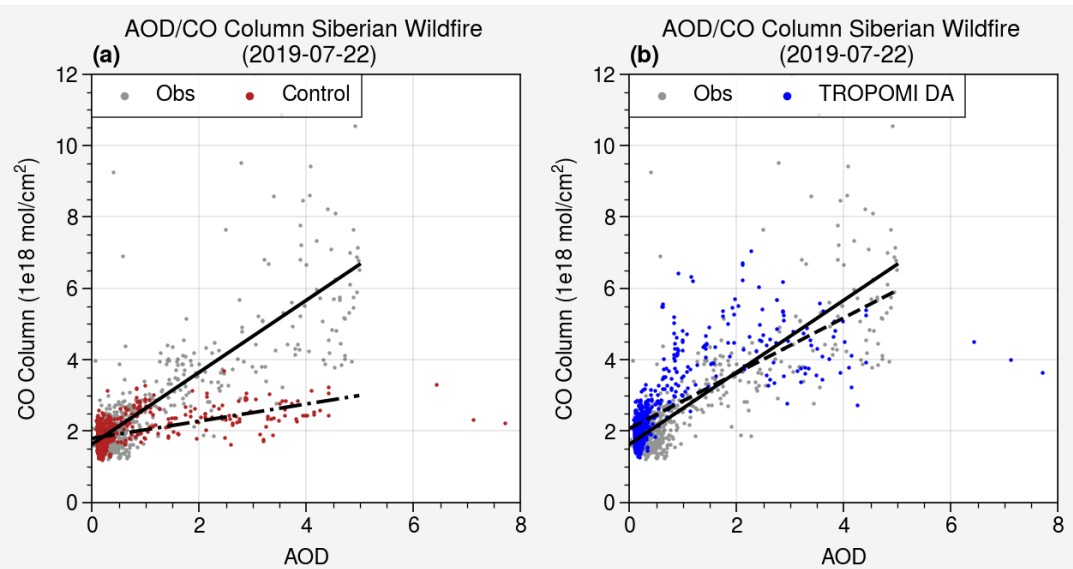

**Figure 13. Linear relationship between AOD and CO column in Siberian wildfire smoke (90°E -150°E, 50°N - 70°N) on 22 July 2019. UFS-RAQMS control (a, red) and UFS-RAQMS TROPOMI CO DA (b, blue) AOD/CO relationships are compared to observed VIIRS AOD/TROPOMI CO (grey).**

**Table 2. Linear relationship between AOD and CO in Siberian wildfire smoke (90°E -150°E, 50°N - 70°N) on 22 July 2019.**

|  | slope | intercept | r |
|---|---|---|---|
| VIIRS AOD/TROPOMI CO Column | 1.0092 | 1.629 | 0.8043 |
| UFS-RAQMS Control AOD/CO Column | 0.2407 | 1.7948 | 0.5648 |
| UFS-RAQMS TROPOMI CO DA AOD/CO Column | 0.7749 | 2.0724 | 0.7106 |

**4.2 Case Study: 16 September 2019 Indonesian Smoke**

During September 2019 wildfire activity over Indonesia contributed to an extreme AOD enhancement in the region. We evaluate binned AOD and CO column on 16 September 2019 for the region 100°E -130°E, 15°S - 15°N.

The spatial distributions of AOD and CO over Indonesia on 16 September 2019 are shown in figure 14 for VIIRS, TROPOMI, the UFS-RAQMS control, and the UFS-RAQMS TROPOMI CO DA experiment. UFS-RAQMS significantly underestimates AOD enhancements in this region, as evident in the Borneo smoke plume and over China. As a result, we also show the UFS-RAQMS AOD scaled by a factor of 3. CO column is significantly underestimated over the maritime continent in UFS-RAQMS control. Agreement with the TROPOMI observations is significantly improved in UFS-RAQMS TROPOMI CO DA.



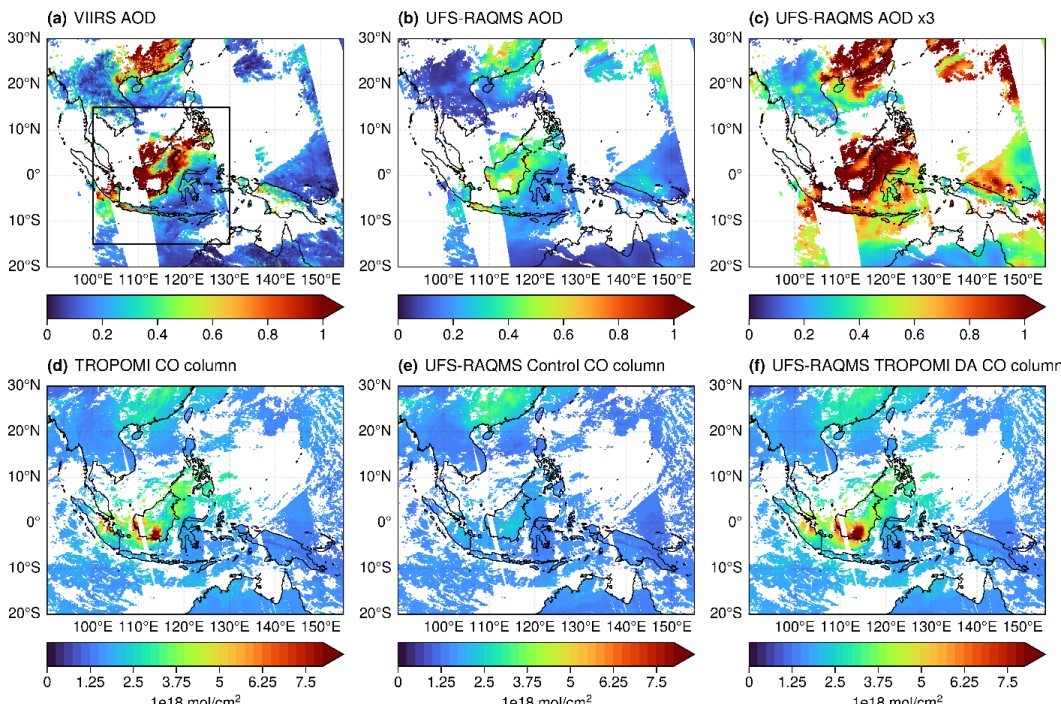

**Figure 14. 16 September 2019 AOD and CO columns over SE Asia. VIIRS AOD (a), UFS-RAQMS AOD (b), UFS-RAQMS AOD scaled by 3 (c), TROPOMI CO column (d), and UFS-RAQMS control (e) and TROPOMI CO DA (f) CO column. Black box in panel a defines region (100°E -130°E, 15°S - 15°N) for AOD/CO column relationship analysis.**

Scatterplots illustrating the relationship between AOD and CO column in Indonesian wildfire smoke are shown in figure 15 for the observations (grey), UFS-RAQMS control (red), and UFS-RAQMS TROPOMI CO DA (blue). The linear regressions are summarized in table 3. VIIRS AOD and TROPOMI CO Column exhibit a compact linear relationship with a slope near 1 and correlation of 0.782. UFS-RAQMS control CO column and AOD are moderately correlated (0.4886), and the slope of the linear relationship is 0.7638, however neither the AOD or CO columns capture

the observed high values. TROPOMI CO DA improves the correlation between AOD and CO Column to 0.7085 but due to the low bias in UFS-RAQMS AOD over the region significantly overestimates the slope of the relationship. To approximate the modeled AOD/CO relationship without the low AOD bias, we apply a scaling factor of 3 to the UFS-RAQMS AOD. Applying this scaling inflates UFS-RAQMS AOD enhancements over Borneo to be closer to observed values (fig 14c, fig. 15c,d). By accounting for the low AOD bias in this way, we obtain a slope for UFS-RAQMS

control CO column and scaled AOD of 0.2546 and for UFS-RAQMS TROPOMI CO DA a slope of 1.275. This points to the need to also assimilate AOD data to improve the agreement with observations in this region.



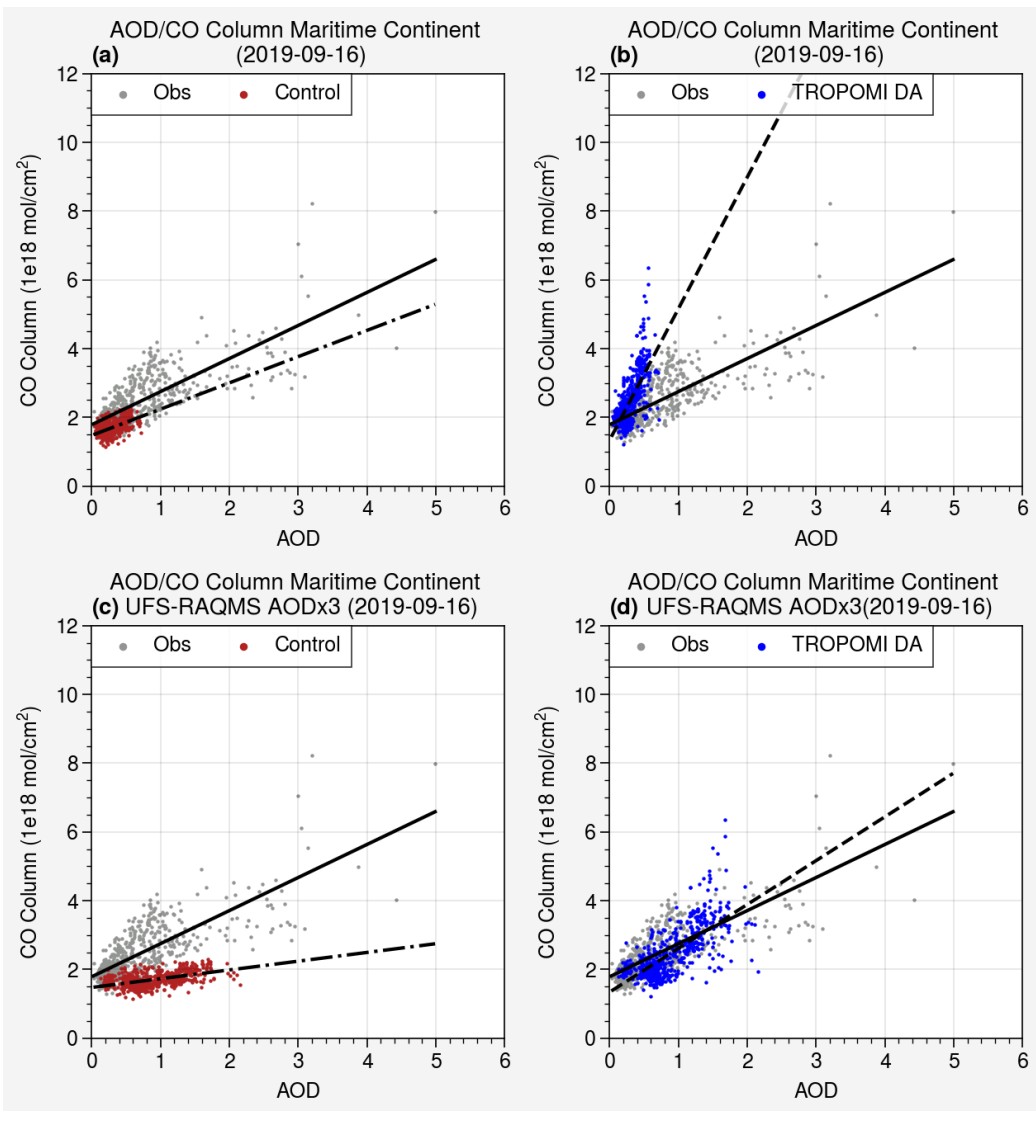

**Figure 15.** Linear relationship between AOD and CO column in Indonesian wildfire smoke (100°E -130°E, 15°S - 15°N) on 16 September 2019. UFS-RAQMS control (a, red) and UFS-RAQMS TROPOMI CO DA (b, blue) AOD/CO relationships are compared to observed VIIRS AOD/TROPOMI CO (grey). UFS-RAQMS control (c, red) and UFS-RAQMS TROPOMI CO DA (d, blue) AODx3/CO relationships are compared to observed VIIRS AOD/TROPOMI CO (grey).




 **Table 3. Linear relationship between AOD and CO column in Indonesian wildfire smoke (100°E -130°E, 15°S - 15°N) on 16 September 2019.**

|  | slope | intercept | r |
|---|---|---|---|
| VIIRS AOD/TROPOMI CO Column | 0.962 | 1.7872 | 0.782 |
| UFS-RAQMS Control AOD/CO Column | 0.7638 | 1.4755 | 0.4886 |
| UFS-RAQMS TROPOMI CO DA AODx3/CO Column | 3.8251 | 1.3404 | 0.7085 |
| UFS-RAQMS Control AODx3/CO Column | 0.2546 | 1.4755 | 0.4886 |
| UFS-RAQMS TROPOMI CO DA AOD/CO Column | 1.275 | 1.3404 | 0.7085 |

**5. Conclusions**

The UFS-RAQMS control experiment significantly underestimates CO column relative to MOPITT and TROPOMI CO column observations. Assimilating TROPOMI CO within UFS-RAQMS using the GSI 3D-var and blended BEC

generally resulted in improved UFS-RAQMS CO analyses relative to satellite, ground-based, and airborne observations. Application of TROPOMI CO DA decreases the average RMSE in CO Column relative to MOPITT and improves correlation between UFS-RAQMS and MOPITT within the FIREX-AQ and CAMP[2]EX domains. TROPOMI CO DA results in an improved CO profile in the free troposphere at most NDACC sites but does increase surface CO biases at high latitude locations and complexity in the vertical structure at many sites. This is a consequence

of using a total column measurement to constrain a profile. Our DA system is minimizing the difference between the TROPOMI observations and the UFS-RAQMS first guess. While the CO column is well constrained, as indicated by the good agreement between UFS-RAQMS TROPOMI CO DA CO columns and MOPITT CO columns, the DA system distributes the analysis increment vertically based on model blended BEC statistics and knowledge of observation errors and vertical sensitivities. Our evaluations with NDACC FTIR CO observations and with field

campaign observations show that this can lead to an over-adjustment near the surface and only small adjustments at high altitudes.

TROPOMI CO DA has the largest impacts in the lower troposphere over Siberia and Indonesia. Our case studies of the relationship between AOD and CO over these regions show that in UFS-RAQMS biomass burning signatures in CO column are not consistent with those in AOD near the biomass burning source regions. Assimilating TROPOMI

CO improves the representation of the biomass burning AOD/CO relationship. We believe this is an indication that the GBBEPx biomass burning CO emissions in UFS-RAQMS are too low. GBBEPx adds biomass burning emissions from VIIRS to the Quick Fire Emissions Database (QFED) biomass burning emissions estimates from MODIS (Zhang et al., 2019). QFED biomass burning aerosol emissions are scaled with biome-representative scale factors for tropical forests, extratropical forests, savanna, and grasslands that were obtained by calibrating NASA Goddard Earth

Observing System Model (GEOS) AOD forecasts with MODIS AOD (Darmenov and da Silva, 2015).

While assimilating CO does compensate for uncertainties in the biomass burning emissions, it does not adjust the biomass burning CO emissions themselves. Since UFS-RAQMS uses emission factors for co-emitted NOx and VOC



species that are based on the GBBEPx biomass burning CO emissions, we anticipate similar uncertainties in these co-emitted species. Future efforts should focus on developing capabilities to use TROPOMI CO column measurements

to adjust the GBBEPx CO biomass burning emissions within UFS-RAQMS. Similar capabilities have been developed using TROPOMI NO2 retrievals to adjust anthropogenic NOx emissions using off-line iterative mass balance approaches (East et al, 2022) and local ensemble transform Kalman filter (LETKF) techniques (Sekiya et al, 2022).

**Code availability**

The version of UFS-RAQMS used to produce the results in this paper is archived on Zenodo

(https://doi.org/10.5281/zenodo.13910346). The version of GSI-RAQMS, used to produce the results in this paper is also archived on Zenodo (https://doi.org/10.5281/zenodo.13905858).

**Data availability**

The FIREX-AQ field campaign data used in this study is publicly available at **http://doi.org/10.5067/SUBORBITAL/FIREXAQ2019/DATA001**. The CAMP²Ex data used in this study is

publicly available at **https://doi.org/10.5067/Suborbital/CAMP2EX2018/DATA001**.
 The NDACC FTIR CO data used in this publication were obtained from James W. Hannigan, Maria V. Makarova, Martine De Mazière, and Nicholas Jones as part of the Network for the Detection of Atmospheric Composition Change (NDACC) and are available through the NDACC website (**www.ndacc.org**).

**Author contributions**

MB and BP conceptualized the study. MB executed the UFS-RAQMS experiments, analyzed the results, and wrote the paper. BP supervised the project and edited the paper. AL developed UFS-RAQMS and the TROPOMI CO data assimilation capability. GD and JD provided the CAMP²Ex and FIREX-AQ CO data. MDM, NJ, and MM provided the NDACC FTIR CO data.

**Competing interests** None

**Acknowledgements**

This research has been supported by the National Oceanic and Atmospheric Administration (grant no. NA20NES4320003).
Thanks to the colleagues at BIRA-IASB and the Université de La Réunion for supporting the FTIR experiment at La Réunion, as well as to the Belgian funding authorities, and to the European Copernicus programme and the Région

Réunion, CNRS and Université de la Réunion for additional financial support.



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
