# Peer review of "UFS-RAQMS Global Atmospheric Composition Model"

_EGUsphere, 2024_

## Author Response (AR1)

Title: UFS-RAQMS Global Atmospheric Composition Model: TROPOMI CO Column Assimilation

Author(s): Maggie Bruckner et al.

MS No.: egusphere-2024-2501

MS type: Development and technical paper

We appreciate the comments made by both reviewers and have revised the manuscript to address the specific comments and suggestions.

The following is the reviewer's comments and our response to each comment in blue.

**Response to Referee #1**

I am just having a hard time following the discussion about aerosols. You mention that sulfate aerosol formation will change with OH. Does the CO assimilation impacts aerosol formation and are the simulated AOD different between the control and the TROPOMI DA? What about secondary organic aerosols and heterogeneous chemistry? This is important to clarify as these should be pathways AOD would change following CO assimilation (without optimizing emissions or transport). Specifically, if AOD are not changed following TROPOMI CO, does it mean that the aerosols are well simulated but not CO (on figure 13)?

The assimilation of TROPOMI CO did not result in significant changes to UFS-RAQMS AOD. We calculate that the largest changes in sulfate AOD and sulfate concentrations on 16 September 2019 to be 5-10% and in airmasses with sulfate AOD < 0.2 and low sulfate concentrations where small changes will have an outsized impact. In the regions with higher AOD and sulfate, the difference between the two UFS-RAQMS experiments for this date is < 2.5%. This discussion has been added to Section 4 of the manuscript in paragraph 2.

The model uses the GOCART aerosol module, which doesn't predict secondary organic aerosols or include heterogeneous chemistry so these pathways are not able to change AOD within the UFS-RAQMS modeling system.

Regarding Figure 14, I don't really follow the practice of increasing AOD by a factor of 3, why don't you just scale the emissions of biomass burning CO and aerosols, using the information from the CO assimilation, even a rough estimate and run forward simulations? Since you are using the same biomass burning emission inputs, AOD and CO errors must be correlated.

The scaling of AOD by a factor of 3 was intended as a rough correction for the underestimation of biomass burning aerosol emissions in this particular region in order to highlight that the poor simulation of the observed AOD/CO ratio is linked to both aerosols and CO. Applying a blanket scaling to the biomass burning emissions would not account for the spatial heterogeneity in the biomass burning emissions underestimation. In response to the reviewer's point that the application of this scaling in our data analysis is somewhat convoluted, we have edited the text and associated figures by dropping the UFS-RAQMS AODx3.

Similarly, with the field campaigns, you have the opportunity to evaluate the impact on other species and assess potential improvements.

We appreciate this suggestion from the reviewer. However, the focus of this paper is to introduce UFS-RAQMS with TROPOMI CO data assimilation and assessing potential improvements beyond deficiencies in biomass burning emissions is beyond the intended scope.

The FTIR observations from NDACC uses a different remote sensing approach than TROPOMI to retrieve CO, but I think it is somehow misleading to call them profiles without caution. I would mention the degrees of freedom for signal, which are probably high enough to allow for a retrieval of a partial tropospheric column at best. If you are willing to keep the figures with the profiles, I would also show the NDACC prior and add appropriate disclaimers.

The NDACC prior and averaging kernels for each site have been added as supplemental figures and the text in the manuscript has been updated to acknowledge the appropriate disclaimers. Mean degrees of Freedom have been added to Table 1 as well.

Minor comments:

L33 please define the TROPOMI acronym and reference here (from line 137: Tropospheric Monitoring Instrument (TROPOMI) (Veefkind et al., 2012)).

The definition of TROPOMI acronym and the reference have been moved to section 1.

L139: define the acronyms UV-near IR and shortwave IR

Definition added for UV and IR acronyms.

L145 (paragraph): Super observations are used to match the model grid's spacing, to reduce noise and avoid overfitting observations, as well as for computational efficiency. Sekiya et al., (2021) found that the smoothing of the spatial variability in analysis increments with OMI NO2 assimilation and noted that the super observations were reducing the number of assimilated observations by a factor of 10, making the use of super-observation to be more relevant with TROPOMI. Note also that the smoothing is actually a desired effect by reducing noise and removing the sub-grid variability that the model cannot represent. I understand there might be a loss of signal while producing the super-observations, so it would have been desirable to actually perform the experiment. You would have been able to show whether the statement line 150 (an underestimates in localized CO column enhancements) ends up being correct or not.

We appreciate this recommendation from the reviewer. However, at this time we are unable to conduct such an experiment due to time constraints and computing resources. It is also beyond the focus of the paper, which is to introduce UFS-RAQMS with TROPOMI CO DA capabilities. We have edited the paragraph to acknowledge the reduced computational cost and the impact on representation error as well as relax the statement in line 150.

L253 and L275: the precision of the Picarro instrument is about 1 ppb for 1-min averages. So, it would be more appropriate to round the biases to significant figures of 1 ppb or 0.1 ppb

Figures and text using aircraft CO measurements have been adjusted to present less precision in bias numbers.

Figure 5: MOPITT retrievals include the dry-air atmospheric column, and it is making it easy to convert both model and observations to XCO. You could also show the bias for Control and TROPOMI DA on panels e) and f).

While it is common in the greenhouse gas community to use XCO, the air quality community typically presents column density (eg. Buchholz et al., 2017). Panels showing the bias have been added to Figure 5.

Buchholz, R. R., Deeter, M. N., Worden, H. M., Gille, J., Edwards, D. P., Hannigan, J. W., Jones, N. B., Paton-Walsh, C., Griffith, D. W. T., Smale, D., Robinson, J., Strong, K., Conway, S., Sussmann, R., Hase, F., Blumenstock, T., Mahieu, E., and Langerock, B.: Validation of MOPITT carbon monoxide using ground-based Fourier transform infrared spectrometer data from NDACC, Atmospheric Measurement Techniques, 10, 1927-1956, 10.5194/amt-10-1927-2017, 2017.

**Response to Referee #2**

General comments:

The manuscript "UFS-RAQMS Global Atmospheric Composition Model: TROPOMI CO Column Assimilation" written by Maggie Bruckner et al. develops the framework of TROPOMI CO data assimilation with the UFS-RAQMS model and 3D-Var, and evaluates its performance against independent satellite, aircraft-campaign, and ground-based remote sensing observations. The manuscript demonstrates the advantages of assimilating TROPOMI CO over the model simulation. This manuscript is generally well written, organized, and designed. I recommend inviting the authors to revise their manuscript to address specific points before a final decision is reached. I provide several comments that need to be addressed before moving forward publication process below:

1. Authors newly construct the background error covariance (BEC) matrix by blending the NMC method and the sensitivity simulation with reduced biomass burning emissions. I would suggest adding sensitivity calculation using (1) blended BEC, (2) NMC BEC, and (3) BB emission BEC. It would be helpful to demonstrate and understand the advantage of the blended BEC approach for readers.

We are unable at this time to perform additional data assimilation experiments using each of the background error statistics. However, we have added a supplemental figure showing the BEC statistics from the NMC method and emission perturbation method. Figure S1 shows that standard deviation is increased near the surface at latitudes with significant biomass burning and that accounting for biomass burning emissions sensitivity increases horizontal length scale in the tropics in approximately the mid-troposphere between model levels 15 and 25. We believe that one of the major uncertainties in the model CO to be emissions, which are better accounted for with the blended BEC. The CO distribution is highly dependent on local emission sources, particularly wildfires as evident in our evaluation of the UFS-RAQMS control experiment (Figure 3 and Figure 5c).

2. Authors state that the UFS-RAQMS experiments include assimilation of MODIS AOD in the methodology section. However, the experiments largely underestimate the VIIRS AOD by a factor of 3. What is the reason why such large negative biases remained even though AOD was assimilated? Also, supposing that these negative biases are attributed to biomass burning emissions, it is more appropriate to apply a scaling factor of 3 to the biomass burning emissions than to the UFS-RAQMS AOD fields.

The MODIS AOD assimilation noted in the methodology section is only applied for the initial conditions used by UFS-RAQMS and does not constrain evolution of the UFS-RAQMS AOD predictions. We have modified the statement about MODIS, OMI, and MLS assimilation in a way that hopefully clarifies this.

We have removed the application of the scaling factor to the UFS-RAQMS AOD fields in section 4.2. The scaling of AOD by a factor of 3 was intended as a rough correction for the underestimation of biomass burning aerosol emissions in maritime continent in order to highlight that the poor simulation of the observed AOD/CO ratio is linked to both aerosols and CO. Future work will look at scaling and adjustment of the biomass burning emissions for both aerosols and CO.

Specific comments:

p. 1, 1. 24—25: What implications the improvements in biomass burning AOD/CO relationships by TROPOMI CO assimilation provide?

The referenced section has been modified and combined with the next line to read:

"Assimilation of TROPOMI CO improves the representation of the biomass burning AOD/CO relationship in UFS-RAQMS by increasing the CO column, which indicates that the biomass burning CO emissions from the Blended Global Biomass Burning Emissions Product (GBBEPx) used in UFS-RAQMS are too low for boreal wildfires."

p. 2, l. 44 "\_CTM forecast vary ...\_": Please clarify if this sentence means the CTM CO levels vary or the CTM forecast performance vary.

We have clarified the sentence to read "CTM concentration fields will vary significantly depending on which biomass burning emission inventory is used".

p. 2, l. 45-48: Is the scheme with the ratio relative to CO for determining the release of other species commonly used? If so, please cite the relevant literature.

Citations have been added for other studies that use emission factors relative to CO for determining the release of other species emitted by biomass burning.

p. 2, l. 51 " CTM fields ": Does it mean concentration fields?

"CTM fields" has been modified to "CTM concentration fields" in order to clarify that chemical DA constrains chemical concentrations.

p. 2, 1. 52–54: Chemical DA methods include ensemble Kalman filter approach in addition to the methods authors mentioned.

The discussion has been updated it to include the ensemble Kalman filter approach.

p. 2, l. 61–62: Please explicitly state what is the advantage of applying TROPOMI CO DA to UFS-RAQMS over previous studies and/or previous version of RAQMS.

We have added the following statements regarding the advantages of UFS-RAQMS with TROPOMI CO DA:

"The application of TROPOMI CO DA provides an observational constraint on model CO concentrations. The previous version of RAQMS utilized the UW hybrid model (Schaack et al., 2004) as the dynamical core and physics parameterizations from the NCAR Community Climate Model (CCM3) (Kiehl et al., 1998). Incorporation of the UFS dynamical core within the new model version updates the physical parameterizations to the suite used to produce operational NOAA forecasts."

p. 4, l. 114–116: What product version do authors used for MODIS AOD, OMI total ozone column, and MLS ozone profiles?

The text has been updated to reflect that for MODIS AOD we use Collection 6.1 (C61), for OMI total column ozone we use the TOMS V8, for MLS stratospheric profiles we used V2. As a note, these products are only assimilated in RAQMS to provide chemical initial conditions and do not directly constrain the UFS-RAQMS experiments beyond the initial conditions.

p. 5, l. 131–135: How did authors determine the inflation factor for standard deviation and the model levels at which two BEC estimates are blended?

The text has been updated as follows to reflect the selection of the inflation factor and the blended model levels.

"This inflation factor was tested during the development of assimilation capabilities within the RAQMS Aura Reanalysis (Bruckner et al., 2024). It accounts for the fact that the 20% emission perturbation used for the emission

sensitivity significantly underestimates the true uncertainties in emissions, which can be an order of magnitude for biomass burning emissions (Al-Saadi et al., 2008). The two BEC estimates are linearly blended in the mid-troposphere (between model levels 15 and 25)."

p. 5, 1.147–149: How do authors account for spatial representativeness errors in the observation errors during the analysis steps?

The observation error uses the quoted biases from the literature and does not account for spatial representativeness through super observations

p. 7, l. 181–183: It might be helpful to show analysis increments for discussion on transport impacts after the data assimilation.

Analysis increments have been added to figure 4.

p. 11, l. 247: Please clarify how to define in-plume measurements.

We have added the clarification that the in-plume measurements from FIREX-AQ are defined using the smoke flag provided in the dataset. The smoke flag is based on enhancements of CO and BC aerosol.

p. 13 and 15, Figures 8 and 10: How does the blended BEC approach affect the improvements in CO vertical profiles?

We cannot quantify how the blended BEC approach affects the improvements in CO vertical profiles since we have not conducted experiments with just the NMC method. We feel that this comparison would be beyond the scope of the study. However, we have added supplemental figures showing the different BEC. p. 16, l. 449—450 "\_We only assimilated the profiles at 700 hPa and the vertical localization reduces the impact towards the surface.\_": It should be described in the method section.

We do not think this comment refers to our manuscript as we were not able to find the referenced line. The data assimilation presented in this work is of total column data and not profile data.

p. 16, l. 490—493: What implication does this contrast btw 500 and 700 hPa provide for model and observation errors?

We do not think this comment refers to our work. The referenced lines are not present in this manuscript and our discussions of model differences from observed profiles refer to altitude in km above sea level rather than in pressure.

p. 16, l. 495—496: What about posterior emissions from MOPITT-DA? I'm curious about CrIS impacts on emissions. Could you show it in the main text or supporting information?

We do not think this comment refers to our work. The referenced lines are not present in this manuscript. While we compare to MOPITT observations, we do not assimilate them nor do we generate posterior emissions.

p. 16, Table 1: I would suggest adding statistics such as mean bias to Table 1 for readability.

The mean bias in the UFS-RAQMS experiments relative to NDACC observations below 25km has been added to the table.